# Exploring Linguistic Properties of Monolingual BERTs with Typological Classification among Languages

**Elena Sofia Ruzzetti**[*,1], **Federico Ranaldi**[*,1], **Felicia Logozzo**[2]
**Michele Mastromattei**[3,1], **Leonardo Ranaldi**[4,1], **Fabio Massimo Zanzotto**[1]
[1]University of Rome Tor Vergata, Italy    [2]University for foreigners of Siena, Italy
[3]Campus Bio-Medico University of Rome, Italy    [4]Idiap Research Institute, Switzerland
elena.sofia.ruzzetti@uniroma2.it    federico.ranaldi@alumni.uniroma2.eu
fabio.massimo.zanzotto@uniroma2.it

## Abstract

The impressive achievements of transformers force NLP researchers to delve into how these models represent the underlying structure of natural language. In this paper, we propose a novel standpoint to investigate the above issue: using typological similarities among languages to observe how their respective monolingual models encode structural information. We aim to layer-wise compare transformers for typologically similar languages to observe whether these similarities emerge for particular layers. For this investigation, we propose to use Centered Kernel Alignment to measure similarity among weight matrices. We found that syntactic typological similarity is consistent with the similarity between the weights in the middle layers, which are the pretrained BERT layers to which syntax encoding is generally attributed. Moreover, we observe that a domain adaptation on semantically equivalent texts enhances this similarity among weight matrices.

## 1 Introduction

Natural language processing (NLP) is dominated by powerful but opaque deep neural networks. The rationale behind this trend is that deep architecture training allows rules and structural information about language to emerge directly from sentences in the target language, sacrificing the interpretable and transparent definition of language regularities. Some exceptions exist where structural syntactic information is explicitly encoded in multilayer perceptrons (Zanzotto et al., 2020) with relevant results on unseen sentences (Onorati et al., 2023). Yet, pre-trained transformers (Vaswani et al., 2017; Devlin et al., 2019) are offered as versatile universal sentence/text encoders that contain whatever is needed to solve any downstream task. These models outperform all other models nearly consistently after fine-tuning or domain-adaptation (Jin et al.,

2022). However, there is no guarantee that when considering languages that share similar structures, the models for those languages will represent those structures in the same way.

Conversely, decades of studies in NLP have created symbol-empowered architectures where everything is explicitly represented. These architectures implement different levels of linguistic analysis: morphology, syntax, semantics, and pragmatics are a few subdisciplines of linguistics that shaped how symbolic-based NLP has been conceived. In this case, a linguistic model of a language – a set of rules and regularities defining its behavior – directly influences the system processing that language. Understanding whether linguistic models emerge in opaque pre-trained transformer architecture is a compelling issue.

*Probing* transformers have mainly been used to investigate whether these model classical linguistic properties of languages. Probing consists of preparing precise sets of examples – probe tasks – and, eventually, observing how these examples activate transformers. In this way, BERT (Devlin et al., 2019) contextual representations have been tested to assess their ability to model syntactic information and morphology (Tenney et al., 2019; Goldberg, 2019; Hewitt and Manning, 2019; Jawahar et al., 2019; Coenen et al., 2019; Edmiston, 2020), also comparing different monolingual BERT models (Nikolaev and Pado, 2022; Otmakhova et al., 2022).

In this study, we take a different standpoint to investigate if traces of linguistic models are encoded in monolingual BERT models: using linguistically-motivated typological similarities among languages (Dryer and Haspelmath, 2013), we aim to layer-wise compare transformers for different languages to observe whether these similarities emerge between weight matrices for particular layers. For this investigation, we propose to use Centered Kernel Alignment to measure similarity among weight

---

*These authors contributed equally to this work

| | |
|---|---|
| Latin: | Hostes (S - 'Enemies') castra (O – 'the camp') oppugnant (V – 'attack') |
| Japanese: | Maria-san (S – 'Mary') wa (particle) sarada (O – 'salad') wo (particle) tabemasu (V – 'eats') |
| Turkish: | Maria (S – 'Mary') elmayi (O – 'the apple') yiyor (V – 'eats') |
| Italian: | Maria 'Maria' (S) mangia 'eats' (V) la mela 'the apple' (O) ' |

Table 1: Syntactic property of languages: the basic order of constituents - subject (S), direct object (O) and verb (V) - in a clause

matrices (Kornblith et al., 2019). We discovered that syntactic typological similarity is consistent with the similarity among weights in the middle layers both in pretrained models and in domain-adapted models after performing domain adaption on a parallel corpus.

## 2 Background and related work

Although different, natural languages show more or less evident similarities at all levels of analysis - phonetics, morphology, syntax, and semantics - due to the fact that "human language capacity is a species-specific biological property, essentially unique to humans, invariant among human groups, and dissociated from other cognitive systems" (Chomsky, 2017). Hence, some properties, called "language universals", are shared by all natural languages: for example, all spoken languages use vowels, all languages have nouns and verbs. Some properties, on the other hand, are common among groups of languages, and are called "universal tendencies": for instance, many languages have nasal consonants. (Greenberg, 2005; Haspelmath, 2001a,b). A classification of languages enables us to organize the shared characteristics of languages.

Languages are classified in two main ways: typological classification and genealogical classification. Language typology is the branch of linguistics that, studying universal tendencies of languages (Comrie, 1989; Song, 2010), maps out "the variation space filled by the languages of the world" and finds "regular patterns and limits of variation" (Haspelmath, 2001a). These typological patterns can be used to classify all languages at different levels of linguistic analysis. Therefore, a typological classification enhances the shared structures among languages.

Typological classification differs from the genealogical (or genetic) one, which aims to categorize languages originating from a common ancestor

(Lehmann, 2013; Dunn, 2015). Indeed, languages belonging to a genealogical family can share specific typological properties or not. Likewise, languages that appear similar, according to a specific typological category, can be genealogically linked or not. For example, Latin, Japanese and Turkish are final-verb languages, whereas Italian is not, despite being directly derived from Latin (see examples in Table 1).

In this work, we focus on typological classification because it directly describes languages as a set of rules of composition of words from the syntactic point of view and morphemes in the morphological analysis. In fact, in the following analysis, our aim is to understand whether rules shared across similar languages are encoded into the corresponding models.

Typological similarities between languages have been previously investigated with probes in two ways: (1) by using syntactic probes in comparing the behavior of different monolingual BERT models (Nikolaev and Pado, 2022; Otmakhova et al., 2022); and, (2) by searching shared syntax representation in multilingual BERT (mBERT) (Chi et al., 2020; Ravishankar et al., 2019; Singh et al., 2019). Chi et al. (2020) suggest that mBERT has a behavior similar to monolingual BERT since its shared syntax representation can be found in middle layers. However, mBERT seems to separate the representations for each language into different subspaces (Singh et al., 2019). Hence, mBERT is not exploiting typological similarities among languages to build better language models: this model does not exploit similarity among languages to reuse part of the representation it builds across different languages.

Another perspective to exploit similarity among languages is to compare activation matrices of transformers produced over probing examples. In this case, the underlying assumption is that parallel sentences should have similar activation matrices if languages are similar. To compare these activation matrices, Canonical Correlation Analysis (CCA) (Hardoon et al., 2004), Singular Vector CCA (SVCCA) (Raghu et al., 2017), and Projection Weighted CCA (PWCCA) (Morcos et al., 2018) have been widely used. More recently, Centered Kernel Alignment (CKA) (Kornblith et al., 2019) measure has been proposed to quantify representational similarity. Kornblith et al. (2019) also claims that CCA and SVCCA are less adequate

for assessing similarity between representations because they are invariant to linear transformations. Using these metrics, Vulić et al. (2020) compared activation matrices derived from monolingual models for pairs of translated words: using CKA, it measured the similarity between the representation of words in different languages. They noticed that the similarity between the representations of words is higher for more similar languages. Similarities between mBERT activation matrices have also been used to derive a phylogenetic tree of languages (Singh et al., 2019; Rama et al., 2020).

In our study, building on typological similarities among languages and on the measures to compare matrices, we aim to compare monolingual BERTs by directly evaluating layer-wise the similarities among parameters of BERTs for different languages. To the best of our knowledge, this is the first study using this idea to investigate whether transformers replicate linguistic models without probing tasks.

## 3 Method and Data

If languages are typologically similar for syntactic or for morphological properties, their BERT models should have some similar weight matrices in some layers. This is our main intuition. Then, in this section, we introduce a method to investigate the above intuition. Section 3.1 describes the selected typological classification of languages and how we use it to compute the similarity between languages. Section 3.2 briefly introduces the subject of our study, BERT. Finally, Section 3.3 introduces a novel way, *biCKA*, to compare weight matrices.

### 3.1 Selected typological similarity among languages

To compute morphological and syntactic similarity among languages, we use a typological classification scheme that is used to generate metric feature vectors for languages. The idea of representing languages using feature vectors and then comparing the similarity between these vectors is in line with previous work, such as lang2vec (Littell et al., 2017). In particular, we stem from WALS to gather typological features of languages.

The World Atlas of Language Structures (WALS) (Dryer and Haspelmath, 2013) aims at classifying languages of the world on the basis of typological criteria at all levels of linguistic analysis. Languages in WALS are represented as categorial feature-value vectors. Each feature can assume two or more values [1]. For instance, the `20A Fusion of Selected Inflectional Formatives` assumes 7 possible values: `1. Exclusively concatenative; 2 Exclusively isolating; 3 Exclusively tonal; 4 Tonal/isolating; 5 Tonal/concatenative; 6 Ablaut/concatenative and 7 Isolating/concatenative`. The feature `25B Zero Marking of A and P Arguments` allows only 2 possibilities: `1. Zero-marking and 2. Non-zero marking`.

Since not all world languages are classified according to each feature – as the list of languages analyzed varies from feature to feature – we had to integrate WALS. Specifically, we took into account all the 12 features classified under "Morphology Area" and a selection of syntactic features of WALS (see Tab. 2). Two linguists added the values for features not defined by WALS.

Regarding morphological features in particular, for Dutch, Romanian, and Swedish we added the values of all features except 26A, which was the only one already compiled in WALS for these languages; we also completed the missing features for Italian except 26A and 27A.

In terms of syntactical features, we selected the most important ones relating to word order (Song, 2012) – from 81A to 97A, Tab. 2 – and some representative characteristics related to linguistic negation (143A, 143E, 143F, 144A). The last-mentioned features about negative morphemes and words have also been selected because these are fully specified for target languages. Unlike morphological features, syntactic features are almost fully specified for all target languages. We only integrated 2 features with the following values: `84A` (Italian, Russian, Rumanian: `1 VOX`, Persian: `3 XOV`, Greek: `6 No dominant order`); `93A` (Italian and Dutch = `1 Initial interrogative phrase`). The complete list of categorical vectors for target languages is available in Appendix A.1: these vectors are extracted from WALS and, eventually, integrated by the two linguists.

Once we have defined all the values of the selected WALS features, we can obtain vectors that represent languages to get their similarity. We use categorical WALS vectors for languages to gen-

---

[1]A complete description of each feature and its values can be found on https://wals.info/feature

| WALS ID | Feature |
|---------|---------|
| | *Morphological features* |
| 20A | Fusion of Selected Inflectional Formatives |
| 21A | Exponence of Selected Inflectional Formatives |
| 21B | Exponence of Tense-Aspect-Mood Inflection |
| 22A | Inflectional Synthesis of the Verb |
| 23A | Locus of Marking in the Clause |
| 24A | Locus of Marking in Possessive Noun Phrases |
| 25A | Locus of Marking: Whole-language Typology |
| 25B | Zero Marking of A and P Arguments |
| 26A | Prefixing vs. Suffixing in Inflectional Morphology |
| 27A | Reduplication |
| 28A | Case Syncretism |
| 29A | Syncretism in Verbal Person/Number Marking |
| | *Syntactic features* |
| 81A | Order of Subject, Object and Verb |
| 82A | Order of Subject and Verb |
| 83A | Order of Object and Verb |
| 84A | Order of Object, Oblique, and Verb |
| 85A | Order of Adposition and Noun Phrase |
| 86A | Order of Genitive and Noun |
| 87A | Order of Adjective and Noun |
| 88A | Order of Demonstrative and Noun |
| 92A | Position of Polar Question Particles |
| 93A | Position of Interrogative Phrases in Content Questions |
| 94A | Order of Adverbial Subordinator and Clause |
| 95A | Relationship between the Order of Object and Verb and the Order of Adposition and Noun Phrase |
| 96A | Relationship between the Order of Object and Verb and the Order of Relative Clause and Noun |
| 97A | Relationship between the Order of Object and Verb and the Order of Adjective and Noun |
| 143A | Order of Negative Morpheme and Verb |
| 143E | Preverbal Negative Morphemes |
| 143F | Postverbal Negative Morphemes |
| 144A | Position of Negative Word With Respect to Subject, Object, and Verb |

Table 2: Selected morphological and syntactic features of WALS

erate metric vectors. Our approach is similar to lang2vec (Littell et al., 2017). In particular, each syntactic and morphological pair of feature-value $(f : v)$ is one dimension of this space, which is 1 for a language L if L has the feature $f$ equal to the value $v$. With this coding scheme, languages are represented as boolean vectors.

Thus, the similarity $\sigma(L_1, L_2)$ between two languages $L_1$ and $L_2$ can be computed as the similarity between the language vectors $\vec{L_1}$ and $\vec{L_2}$, using the cosine similarity ($cos$) as similarity measure:

$$\sigma(L_1, L_2) = cos(\vec{L_1}, \vec{L_2})$$

This measure assesses the similarity between languages by counting the number of shared feature-value pairs. We can compute two versions of this measure – $\sigma_{synt}(L_1, L_2)$ and $\sigma_{morph}(L_1, L_2)$ – which respectively compute similarity among languages using syntactic and morphological features. Although this formulation is similar to lang2vec, we encoded the WALS features directly into metric vectors because lang2vec does not contain morphological features and cannot benefit from our integration of WALS for the syntactic feature.

Then, we can assess the typological similarity of the languages as defined by linguists. We can now investigate whether the same similarity holds among different BERT models.

## 3.2 BERT Model in brief

Among the different transformer-based architectures, BERT (Devlin et al., 2019) is the subject of our study since it is a widely known architecture, and many BERT models – in different languages – are available. This section briefly describes how it is organized in order to give names to weight matrices at each layer.

BERT is a layered transformer-based model with attention (Vaswani et al., 2017) that uses only the Encoder block. Each layer of the Encoder block is divided into two sub-layers. The first sub-layer, called Attention Layer, heavily relies on the attention mechanism. The second sub-layer (Feed Forward Layer) is simply a feed-forward neural network. Clearly, each sub-layer has its weight matrices referred to in the rest of the paper as follows:

| Attention | |
|-----------|---|
| query | $Q_i$ |
| key | $K_i$ |
| value | $V_i$ |
| attention output dense | $OA_i$ |
| **Feed Forward Network** | |
| intermediate dense | $DI_i$ |
| output dense | $DO_i$ |

For each monolingual BERT for a language $L$, weight matrices $W_i = \{Q_i, K_i, V_i, OA_i, DI_i, DO_i\}$ for $i \in [0, \ldots, 11]$ should represent part of the linguistic model of $L$ after pre-training.

## 3.3 Bidimensional Centered Kernel Alignment for Comparing Weight Matrices

Centered kernel alignment (CKA) (Kornblith et al., 2019) is a metric used to compare similarities between representations, that is, activation matrices of the deep neural network. Linear CKA measure is defined as follows:

$$CKA(X, Y) = \frac{||Y^T X||_F^2}{||X^T X||_F ||Y^T Y||_F} \quad (1)$$

where $X$ and $Y$ are $n \times p$ matrices and denote activations of $p$ neurons for $n$ examples. The key idea behind this metric is to determine the similarity between pairs of elements and then compare the similarity structures. As presented by Kornblith et al. (2019), computing the numerator in Equation 1 is proportional to computing the similarity, using the dot product between the linearized version (which is referred to as $vec$) of the matrices

as similarity measure, of the estimated covariance matrices $XX^T$ and $YY^T$ for mean-centered activation matrices $X$ and $Y$:

$$vec(XX^T)^T \cdot vec(YY^T) = trace(XX^TYY^T) = ||Y^TX||_F^2 \quad (2)$$

where $vec(A)$ is the linearization of a matrix $A$ and, thus, $vec(XX^T)^T \cdot vec(YY^T)$ is the equivalent of the Frobenius product between two matrices. Then, CKA is suitable to compare matrices of representations of sentences as it is possible to derive relations between rows of the two matrices as these represent words.

Among the different methods, CKA is chosen because it allows one to compare matrices with $p \geq n$. Other metrics that are invariant to invertible linear transformations, such as CCA and SVCCA, instead assign the same similarity value to any pair of matrices having $p \geq n$ (Kornblith et al., 2019). Moreover, the columns of the analyzed weight matrices are characterized by high multicollinearity. If one feature is a linear combination of others, then also the covariance matrix has some row that is linear combination of others and hence is rank deficient. This makes difficult in this setting the use of PWCCA since it requires the computation of CCA and, consequently, the inversion of the covariance matrices.

However, in the case of weight matrices, CKA is not suitable as both rows and columns of these matrices play an important role in determining the output of the network. For example, during the computation of the self-attention in a layer $i$, given $X_{i-1}$ the output of the previous layer, the product between the query activation matrix $X_{i-1}Q_i$ and the key activation matrix (transposed) $X_{i-1}K_i$ is often described as a function of column vectors of $Q_i$ and $K_i$ as $X_{i-1}Q_i(X_{i-1}K_i)^T$. However, the attention mechanism is implicitly computing the product $Q_iK_i^T$ (where rows of $Q_i$ and $K_i$ are compared). Hence, both covariance matrices of rows and columns describe something about the input transformation and need to be studied.

Indeed, given a pair of weight matrices $W_1$ and $W_2$, both the similarity $W_1W_1^T$ with respect to $W_2W_2^T$ and the similarity of $W_1^TW_1$ with respect to $W_2^TW_2$ may play an important role in determining the similarity between the two weight matrices.

Hence, we introduce bidimensional CKA, $biCKA$, a variant of CKA that compares matrices considering rows and columns.

Firstly, given a weight matrix $W$, we define the following block diagonal matrix $F(W)$ as follows:

$$F(W) = \left[ \begin{array}{c|c} W & 0 \\ \hline 0 & W^T \end{array} \right]$$

Then, $biCKA$, our solution to compute the similarity between weight matrices, is defined as follows:

$$biCKA(W_1, W_2) = CKA(F(W_1), F(W_2)))$$

Hence:

$$biCKA(W_1, W_2) \propto$$
$$\propto vec\left( \left[ \begin{array}{c|c} W_1W_1^T & 0 \\ \hline 0 & W_1^TW_1 \end{array} \right] \right)^T \cdot vec\left( \left[ \begin{array}{c|c} W_2W_2^T & 0 \\ \hline 0 & W_2^TW_2 \end{array} \right] \right)$$

$biCKA(W_1, W_2)$ – relying on the normalized similarity between the block covariance matrices of $F(W_1)$ and $F(W_2)$ – takes into account the similarity between rows and columns of weight matrices $W_1$ and $W_2$.

Given a pair of weight matrices, $W_1$ and $W_2$, $biCKA$ is our solution to compute their similarity.

## 4 Experiments

### 4.1 Experimental set-up

The aim of the experiments is to use typological similarity between languages to determine whether some BERT layers encode morphological or syntactic information.

Our working hypothesis is the following: given a set of pairs of languages $(L_1, L_2)$, a particular matrix $W$ of a BERT layer encodes syntactic or morphological information if the similarity $biCKA(W_{L_1}, W_{L_2})$ between that particular weight matrix of $\text{BERT}_{L_1}$ and $\text{BERT}_{L_2}$ correlates with the typological similarities $\sigma_{synt}(L_1, L_2)$ or $\sigma_{morph}(L_1, L_2)$, respectively. To evaluate the correlation, we compared two lists of pairs ranked according to $biCKA(W_{L_1}, W_{L_2})$ and $\sigma(L_1, L_2)$ by using the Spearman's correlation coefficients.

To select languages for the sets of language pairs, we used Hugging Face[2] that offers a considerable repository of pre-trained BERTs for a variety of languages. Hence, we had the possibility to select typologically diverse languages for our investigation. Languages are listed along with the respective pretrained monolingual model (retrieved from the Hugging Face repository using the `transformers` library (Wolf et al., 2020)).

We selected the following European languages, listed above, according to genealogical criteria:

---
[2] https://huggingface.co/models

- Indo-European Romance (or Neo-Latin) languages: Italian (ITA), `bert-base-italian-cased` (Schweter, 2020c); French (FRE), `bert-base-french-europeana-cased` (Schweter, 2020b); Spanish (SPA), `bert-base-spanish-wwm-cased` (Cañete et al., 2020); Romanian (ROM), `bert-base-romanian-cased-v1` (Dumitrescu et al., 2020);

- Indo-European Germanic languages: English (ENG), `bert-base-uncased` (Devlin et al., 2019); Swedish (SWE), `bert-base-swedish-cased` (Malmsten et al., 2020); German (GER), `bert-base-german-cased` (Deepset, 2019); Dutch (DUT), `bert-base-dutch-cased` (de Vries et al., 2019);

- Indo-European Slavic languages: Russian (RUS), `rubert-base-cased` (Kuratov and Arkhipov, 2019);

- Indo-European Hellenic languages: Greek (GRK), `bert-base-greek-uncased-v1` Koutsikakis et al. (2020);

- Non-Indo-European languages: Turkish (TUR), `bert-base-turkish-cased` (Schweter, 2020a); Finnish (FIN), `bert-base-finnish-cased-v1` Virtanen et al. (2019)

In addition, we included Persian (or Farsi) (PRS), belonging to the Indo-Iranian branch with the corresponding model `bert-fa-base-uncased` by Farahani et al. (2020), which is one of the most representative Indo-European language spoken outside the European continent.

Given the above list of languages, we performed three different sets of experiments:

1. FULL - all pairs of languages in the list are considered in the set;

2. FOCUSED - first, we cluster languages in groups and then only pairs of languages from different groups are retained;

3. DOMAIN ADAPTED - on a selected set of language pairs, we perform domain adaption of the models on sentences from a parallel corpus.

The FULL experiments aim to detect where BERT models for typological similar languages may encode syntactic and morphological features, using WALS-based similarity among languages and our novel $biCKA$ among weight matrices.

The FOCUSED experiments on clustering languages aim to be more specific in describing whether some weight matrices encode specific linguistic information. Indeed, despite being all European, selected languages show different typological features. Hence, these languages may be clustered according to their WALS similarity on typological vectors. Then, we identified distinctive features among pairs of clusters by comparing the features Gini impurity (Ceriani and Verme, 2012) of each cluster and their union. Given two clusters, such as $S1$ and $S2$, the most interesting features will be those features that distinguish the two clusters. The impurities of these features in $S1$ and $S2$ are as low as possible and, at the same time, present in each cluster a lower impurity than the impurity measured in the union of clusters $S1 \cup S2$. Among these interesting features, we will refer to features that take entirely different values between clusters as *polarizing features*. Once those features are detected, to understand if they are encoded by a given matrix, to compute ranked lists of language pairs, we selected pairs of languages belonging to different clusters. The intuition is that the similarity between these pairs of languages, and, thus, Spearman's correlation between ranked lists, is mainly affected by polarizing features.

Finally, the last DOMAIN ADAPTED experiment shows how models behave in a setting closer to the ideal one, with shared training procedure and training data. Since data are semantically identical, the only differences are in morphological and typological features, which hence we hypothesize should be represented similarly by models trained on similar texts. Domain adaptation is performed by fine-tuning each model on Masked Language Modeling task. The data used are from the EuroParl corpus (Koehn, 2005), a large-scale parallel corpus of proceedings of the European Parliament. Hence, we focus on languages spoken in the European Union, excluding Turkish, Russian, and Persian in this analysis since no translation is available for these languages. Each model is trained for four epochs on a subset of sentences that has a translation in all languages (50k sentences).

# 5 Results and Discussion

This section reports the results of the three sets of experiments: FULL - all pairs of languages (Sec. 5.1); FOCUSED - only pairs from different clusters (Sec. 5.2); DOMAIN ADAPTED - analysis is performed after domain adaptation on a parallel corpus (Sec. 5.3).

## 5.1 Correlation between Matrices in all Languages: the FULL approach

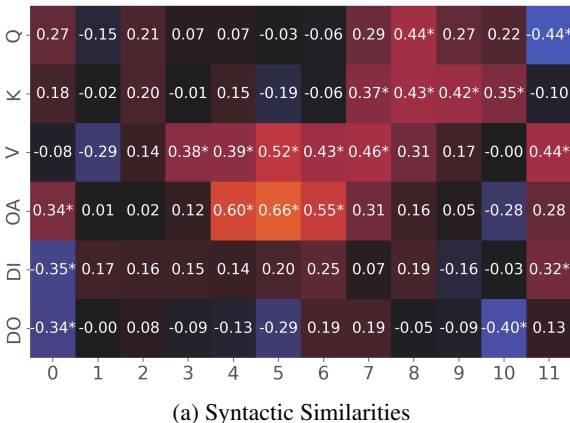

(a) Syntactic Similarities

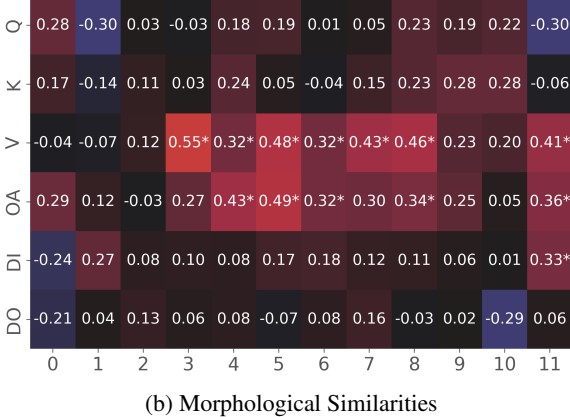

(b) Morphological Similarities

Figure 1: Spearman's correlation coefficient over all language pairs ranked with typological features and with weight matrix similarities. Rows are the matrices types, and columns are the layers. Values closer to $+1$ are in red, closer to 0 are in black, and values closer to $-1$ are in blue. Statistically significant results with a $p - value < 0.01$ are labeled with $^*$.

Experiments in the FULL configuration – using all pairs of languages in a single set – have relevant results showing that typological similar languages led to similar models. In particular, the syntactic spaces in WALS correlate with specific layers in BERT (see Fig. 1).

Syntactic properties seem to emerge in middle layers (Fig. 1a) for content matrices $OA$ and $V$ of the attention sub-layer. In fact, matrices $OA_4$, $V_5$, $OA_5$, $OA_6$ achieve a Spearman's value higher than the fixed threshold, respectively, 0.60, 0.52, 0.66, and 0.55. The higher correlation value is 0.66 for $OA_5$. All Spearman's are tested against the null hypothesis of the absence of linear correlation and are statistically significant with a p-value smaller than 0.01 with a Student t-test. These results align with the probing experiments that assessed the predominance of middle layers in the encoding of syntactic information (Tenney et al., 2019).

Morphological properties instead show up in a single matrix $V_3$. The correlation between typological space and the matrix is moderate, that is, 0.55. Moreover, the correlation seems to be moderately stable until the layer 7 on these matrices $V$.

This analysis confirms our hypothesis that syntactically similar languages induce similar models and that this can be observed directly on weight matrices at layers 4, 5, and 6, as expected since probing tasks detect syntax in middle layers (Tenney et al., 2019).

## 5.2 FOCUSED analysis: measuring Extra-Cluster Correlation

Experiments on correlations of the ranked list of language pairs extracted from different language clusters aim to study whether specific linguistic features are related to specific weight matrices in BERT.

For this set of experiments, we performed a K-means clustering of languages (see Appendix A.2) for both syntactic and morphological typological spaces. Four clusters emerged for the syntactic typological space of languages, and three clusters emerged on the morphological one. The clusters ($S$) generated from syntactic features are: $S1 = \{$ITA, FRE, SPA, ROM$\}$, $S2 = \{$ENG, FIN, SWE, RUS, GRK$\}$, $S3 = \{$TUR, PRS$\}$ and $S4 = \{$GER, DUT$\}$. The clusters ($M$) generated from morphological space are: $M1 = \{$ITA, FRE, SPA, ROM$\}$, $M2 = \{$ENG, SWE, GER, DUT$\}$ and $M3 = \{$TUR, FIN, PRS, RUS, GRK$\}$.

The Spearman's correlation is reported for each matrix in the different layers, for both syntactic (Fig. 2a) and morphological (Fig. 2b) clustering: we focus here on results obtained by comparing the larger clusters since the Spearman's correlations are negatively affected by smaller cluster size. A complete list of correlations can be found in Ap-

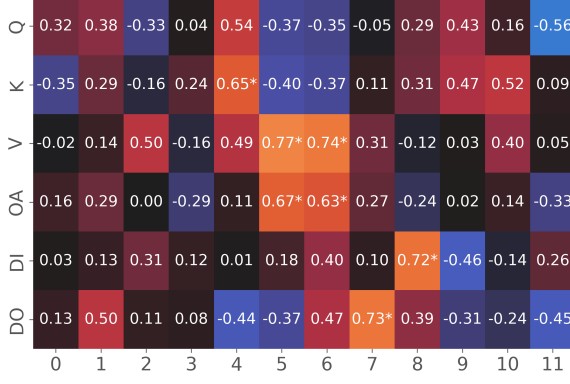

(a) Syntactic Similarities on Clusters $S1$ and $S2$

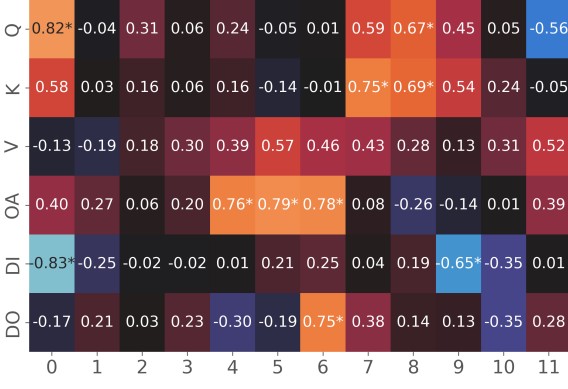

(b) Morphological Similaritie on Clusters $M1$ and $M3$

Figure 2: Each matrix shows Spearman's correlation coefficients for extra-cluster typological similarities and weight matrix similarities. Rows are the matrices types, and columns are the layers. Values closer to $+1$ are in red, closer to 0 are in black, and values closer to -1 are in blue. Statistically significant results with a $p-value < 0.01$ are labeled with $^*$.

pendix A.2.

From the syntactic point of view, the two larger clusters $S1$ and $S2$ show an interesting pattern (see Fig. 2a). The threshold of 0.5 is exceeded by the matrices in layers from 4 to 8, with a peak of 0.77 in layer 5 on matrix $V_5$. These values are statistically significant with a p-value lower than 0.01. Those results confirm what has been observed in the previous section. Hence, these matrices may encode the polarizing features 87A Order of Adjective and Noun and 97A Relationship between the Order of Object and Verb and the Order of Adjective and Noun.

From the morphological point of view (see Fig. 2), language pairs from $M1$-$M3$ lead to interesting results. Investigating the morphological clustering, less regular patterns can be found by looking at Spearman's correlation coefficient.

In fact, Spearman's coefficients are above the threshold across multiple layers and are statistically significant with a peak at layer 0 ($Q_0$ matrix). However, a high correlation can also be observed in other layers, with no clear descending trend. Some of the polarizing and nearly polarizing features are 29A Syncretism in Verbal Person/Number Marking, 21A Exponence of Selected Inflectional Formatives, 23A Locus of Marking in the Clause.

Although these results are not conclusive, these experiments on clusters of similar languages establish a possible new way to investigate the linguistic properties of BERT.

## 5.3 Correlations after Domain Adaptation on Parallel Corpus: the DOMAIN ADAPTED approach

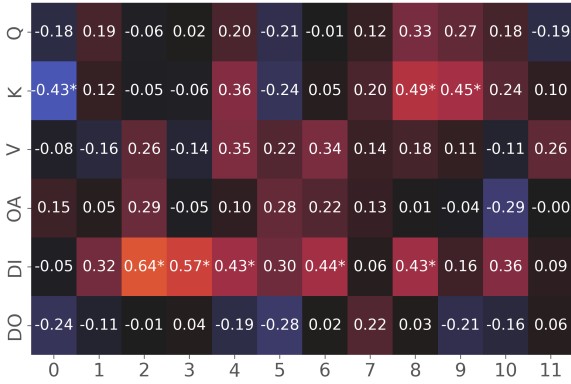

(a) Syntactic Similarities before domain adaptation

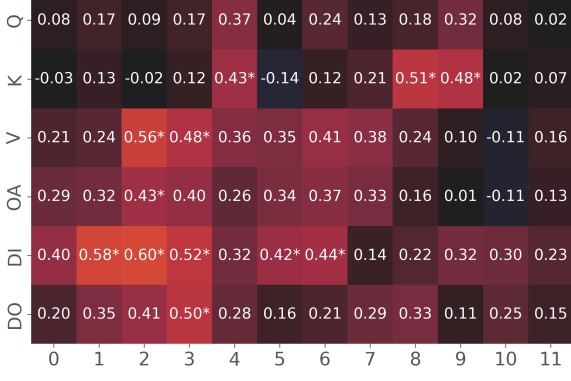

(b) Syntactic Similarities after domain adaptation

Figure 3: Spearman's correlation coefficient over European language pairs ranked with typological features and with weight matrix of domain adapted BERT similarities 3b and of pretrained BERT similarities. Rows are the matrices types, and columns are the layers. Values closer to $+1$ are in red, closer to 0 are in black, and values closer to -1 are in blue. Statistically significant results with a $p-value < 0.01$ are labeled with $^*$.

In this section, we observe Spearman's correlation on a subset of models after domain adaption. As described in Section 4.1, we analyze here models adapted on a parallel corpus with Masked Language Modeling task. When selecting models for the FULL setup, we observed that they were all pretrained on a wide variety of sources and domains. However, dissimilar pretraining data could negatively affect the analyzed correlations. Our hypothesis is that conditioning models' weights on semantically equivalent sentences can lead to a clearer analysis of syntactic similarities across typological similar languages. We focus here on syntactic analysis since it leads to more interesting results in all set of experiments, while morphological analysis for completeness is reported in the Appendix A.3.

Our hypothesis is confirmed by the syntactic analysis experiments since the positive trend is confirmed across the middle layers, and relatively higher statistically significant correlation coefficients can be observed (Figure 3). We can identify a positive trend from layer 2 to layer 6 on matrix $DI$, and on matrices $V$ and $OA$ with smaller yet consistent positive values on the same layers. The average increase in correlation from layer 2 to layer 6 is 0.18. Moreover, the correlations do not tend to increase on the latter layers (especially layers 10 and 11), which are the layers generally associated with semantic tasks.

These experiments give us an important confirmation that the typological similarity across different models, while being present in pretrained models, is enhanced by conditioning weights to semantically equivalent data, which thus differ only syntactically and morphologically.

## 6 Conclusions

Understanding whether large language models encode linguistic models is a challenging and important research question. Despite their flexibility and their SOTA performances on multiple NLP tasks, transformer-based models do not explicitly express what they learn about language constructions and, thus, it is hard to control their failures directly.

Our paper proposes a novel way to observe BERT capability to encode linguistic models and achieves two critical contributions: confirming syntactic probing results from another point of view and opening a new way to investigate specific linguistic features in layers. Differently from all pre-

vious approaches, our methodology is based on directly comparing layer-by-layer weight matrices of BERTs and evaluating whether typological similar languages have similar matrices in specific layers. From a different standpoint, our experimental results confirm that layers 4, 5, and 6 encode syntactic linguistic information. Moreover, they also suggest that the attention's value matrix V and the attention's output layer are more important than other matrices in encoding linguistic models. This latter is an important and novel result. In fact, these findings can be important because they can help interpret the inner workings of these models to go toward the so-called actionable explainability indicating which matrix weights could be changed to obtain a desired behavior. Moreover, this methodology could help to build a more informed architecture that takes advantage of the most promising layers in multilingual models.

Hence, in future work, our findings could be helpful: (1) for defining cross-language training procedures that consider similarities between languages and between models, and (2) for fostering ways to act in specific weight matrices of specific layers of BERT to change the undesired behavior of final BERT downstream systems. Moreover, this methodology could be used on other transformer architectures.

## Limitations

To the best of our knowledge, this is the first attempt to directly compute similarities between weight matrices of BERT and to compare it with an external resource. Hence, it has many possible limitations.

The more important limitation can be due to the fact that transformers, in general, and BERT, in particular, could be mostly large memories of pre-training examples. Hence, comparing weight matrices at different layers could imply comparing pre-training examples given to the different BERTs. However, this is not only a limitation of our study. Indeed, it could be a limitation for any linguistic analysis of BERTs or other transformers.

The second limitation is the availability of monolingual BERTs for low-resource languages, which led to an analysis that is incomplete. The growing availability of monolingual BERTs can solve this issue and may require to re-do the experiments.

The third limitation is the incompleteness of the World Atlas of Language Structures (WALS)

([Dryer and Haspelmath, 2013](#)). Indeed, as this is a growing linguistic resource, our results also depend on the quality of the resource at the time of download. For this reason, we selected languages that may be controlled by our linguists.

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

# A   Appendix

## A.1   WALS and integration of morphological and syntactic features

In this Section we present the complete list of WALS features used in this work. The values listed in the following tables are defined from the WALS website and, where needed, analyzed and inserted from the linguist Felicia Logozzo, as also described in Section 3.1.

| wals_code | 81A "Order of Subject, Object and Verb" | 82A Order of Subject and Verb | 83A Order of Object and Verb | 84A "Order of Object Oblique, and Verb" | 85A Order of Adposition and Noun Phrase | 86A Order of Genitive and Noun |
|---|---|---|---|---|---|---|
| ita | 2 SVO | 3 No dominant order | 2 VO | 1 VOX | 2 Prepositions | 2 Noun-Genitive |
| eng | 2 SVO | 1 SV | 2 VO | 1 VOX | 2 Prepositions | 3 No dominant order |
| tur | 1 SOV | 1 SV | 1 OV | 3 XOV | 1 Postpositions | 1 Genitive-Noun |
| fin | 2 SVO | 1 SV | 2 VO | 1 VOX | 1 Postpositions | 1 Genitive-Noun |
| swe | 2 SVO | 1 SV | 2 VO | 1 VOX | 2 Prepositions | 1 Genitive-Noun |
| prs | 1 SOV | 1 SV | 1 OV | 3 XOV | 2 Prepositions | 2 Noun-Genitive |
| ger | 7 No dominant order | 1 SV | 3 No dominant order | 6 No dominant order | 2 Prepositions | 2 Noun-Genitive |
| fre | 2 SVO | 1 SV | 2 VO | 1 VOX | 2 Prepositions | 2 Noun-Genitive |
| rus | 2 SVO | 1 SV | 2 VO | 1 VOX | 2 Prepositions | 2 Noun-Genitive |
| spa | 2 SVO | 3 No dominant order | 2 VO | 1 VOX | 2 Prepositions | 2 Noun-Genitive |
| dut | 7 No dominant order | 1 SV | 3 No dominant order | 6 No dominant order | 2 Prepositions | 2 Noun-Genitive |
| rom | 2 SVO | 1 SV | 2 VO | 1 VOX | 2 Prepositions | 2 Noun-Genitive |
| grk | 7 No dominant order | 3 No dominant order | 2 VO | 6 No dominant order | 2 Prepositions | 2 Noun-Genitive |

| wals_code | 87A Order of Adjective and Noun | 88A Order of Demonstrative and Noun | 92A Position of Polar Question Particles | 93A Position of Interrogative Phrases in Content Questions | 94A Order of Adverbial Subordinator and Clause | 95A Relationship between the Order of Object and Verb and the Order of Adposition and Noun Phrase |
|---|---|---|---|---|---|---|
| ita | 2 Noun-Adjective | 1 Demonstrative-Noun | 6 No question particle | 1 Initial interrogative phrase | 1 Initial subordinator word | 4 VO and Prepositions |
| eng | 1 Adjective-Noun | 1 Demonstrative-Noun | 6 No question particle | 1 Initial interrogative phrase | 1 Initial subordinator word | 4 VO and Prepositions |
| tun | 1 Adjective-Noun | 1 Demonstrative-Noun | 2 Final | 2 Not initial interrogative phrase | 5 Mixed | 1 OV and Postpositions |
| fin | 1 Adjective-Noun | 1 Demonstrative-Noun | 3 Second position | 1 Initial interrogative phrase | 1 Initial subordinator word | 3 VO and Postpositions |
| swe | 1 Adjective-Noun | 1 Demonstrative-Noun | 6 No question particle | 1 Initial interrogative phrase | 1 Initial subordinator word | 4 VO and Prepositions |
| prs | 2 Noun-Adjective | 1 Demonstrative-Noun | 1 Initial | 2 Not initial interrogative phrase | 1 Initial subordinator word | 2 OV and Prepositions |
| ger | 1 Adjective-Noun | 1 Demonstrative-Noun | 6 No question particle | 1 Initial interrogative phrase | 1 Initial subordinator word | 5 Other |
| fre | 2 Noun-Adjective | 1 Demonstrative-Noun | 1 Initial | 1 Initial interrogative phrase | 1 Initial subordinator word | 4 VO and Prepositions |
| rus | 1 Adjective-Noun | 1 Demonstrative-Noun | 3 Second position | 1 Initial interrogative phrase | 1 Initial subordinator word | 4 VO and Prepositions |
| spa | 2 Noun-Adjective | 1 Demonstrative-Noun | 6 No question particle | 1 Initial interrogative phrase | 1 Initial subordinator word | 4 VO and Prepositions |
| dut | 1 Adjective-Noun | 1 Demonstrative-Noun | 6 No question particle | 1 Initial interrogative phrase | 1 Initial subordinator word | 5 Other |
| rom | 2 Noun-Adjective | 6 Mixed | 6 No question particle | 1 Initial interrogative phrase | 1 Initial subordinator word | 4 VO and Prepositions |
| grk | 1 Adjective-Noun | 1 Demonstrative-Noun | 1 Initial | 1 Initial interrogative phrase | 1 Initial subordinator word | 4 VO and Prepositions |

| wals_code | 96A Relationship between the Order of Object and Verb and the Order of Relative Clause and Noun | 97A Relationship between the Order of Object and Verb and the Order of Adjective and Noun | 143A Order of Negative Morpheme and Verb | 143E Preverbal Negative Morphemes | 143F Postverbal Negative Morphemes | 144A "Position of Negative Word With Respect to Subject, Object, and Verb" |
|---|---|---|---|---|---|---|
| ita | 4 VO and NRel | 4 VO and NAdj | 1 NegV | 1 NegV | 4 None | 2 SNegVO |
| eng | 4 VO and NRel | 3 VO and AdjN | 1 NegV | 1 NegV | 4 None | 2 SNegVO |
| tun | 1 OV and RelN | 1 OV and AdjN | 4 [V-Neg] | 4 None | 2 [V-Neg] | 20 MorphNeg |
| fin | 4 VO and NRel | 3 VO and AdjN | 1 NegV | 1 NegV | 4 None | 2 SNegVO |
| swe | 4 VO and NRel | 3 VO and AdjN | 6 Type 1 / Type 2 | 1 NegV | 1 VNeg | 16 More than one position |
| prs | 2 OV and NRel | 2 OV and NAdj | 3 [Neg-V] | 2 [Neg-V] | 4 None | 20 MorphNeg |
| ger | 5 Other | 5 Other | 6 Type 1 / Type 2 | 1 NegV | 1 VNeg | 16 More than one position |
| fre | 4 VO and NRel | 4 VO and NAdj | 15 OptDoubleNeg | 1 NegV | 1 VNeg | 19 OptDoubleNeg |
| rus | 4 VO and NRel | 3 VO and AdjN | 1 NegV | 1 NegV | 4 None | 2 SNegVO |
| spa | 4 VO and NRel | 4 VO and NAdj | 1 NegV | 1 NegV | 4 None | 2 SNegVO |
| dut | 5 Other | 5 Other | 6 Type 1 / Type 2 | 1 NegV | 1 VNeg | 16 More than one position |
| rom | 4 VO and NRel | 4 VO and NAdj | 1 NegV | 1 NegV | 4 None | 2 SNegVO |
| grk | 4 VO and NRel | 3 VO and AdjN | 1 NegV | 1 NegV | 4 None | 16 More than one position |

| wals_code | 20A Fusion of Selected Inflectional Formatives | 21A Exponence of Selected Inflectional Formatives | 22A Inflectional Synthesis of the Verb | 23A Locus of Marking in the Clause | 24A Locus of Marking in Possessive Noun Phrases | 25A Locus of Marking: Whole-language Typology |
|---|---|---|---|---|---|---|
| ita | 1 Exclusively concatenative | 5 No case | 3 4-5 categories per word | 4 No marking | 2 Dependent marking | 5 Inconsistent or other |
| eng | 1 Exclusively concatenative | 5 No case | 2 2-3 categories per word | 2 Dependent marking | 2 Dependent marking | 2 Dependent-marking |
| tur | 1 Exclusively concatenative | 1 Monoexponential case | 4 6-7 categories per word | 2 Dependent marking | 3 Double marking | 5 Inconsistent or other |
| fin | 1 Exclusively concatenative | 2 Case + number | 2 2-3 categories per word | 2 Dependent marking | 3 Double marking | 5 Inconsistent or other |
| swe | 1 Exclusively concatenative | 5 No case | 2 2-3 categories per word | 4 No marking | 2 Dependent marking | 5 Inconsistent or other |
| prs | 1 Exclusively concatenative | 1 Monoexponential case | 3 4-5 categories per word | 3 Double marking | 1 Head marking | 5 Inconsistent or other |
| ger | 1 Exclusively concatenative | 2 Case + number | 2 2-3 categories per word | 2 Dependent marking | 2 Dependent marking | 2 Dependent-marking |
| fre | 1 Exclusively concatenative | 5 No case | 3 4-5 categories per word | 4 No marking | 2 Dependent marking | 5 Inconsistent or other |
| rus | 1 Exclusively concatenative | 2 Case + number | 3 4-5 categories per word | 2 Dependent marking | 2 Dependent marking | 2 Dependent-marking |
| spa | 1 Exclusively concatenative | 1 Monoexponential case | 3 4-5 categories per word | 3 Double marking | 2 Dependent marking | 5 Inconsistent or other |
| dut | 1 Exclusively concatenative | 5 No case | 2 2-3 categories per word | 4 No marking | 2 Dependent marking | 5 Inconsistent or other |
| rom | 1 Exclusively concatenative | 3 Case + referentiality | 3 4-5 categories per word | 2 Dependent marking | 2 Dependent marking | 2 Dependent-marking |
| grk | 1 Exclusively concatenative | 2 Case + number | 3 4-5 categories per word | 3 Double marking | 3 Double marking | 3 Double-marking |

| wals_code | 26A Prefixing vs. Suffixing in Inflectional Morphology | 27A Reduplication | 28A Case Syncretism | 21B Exponence of Tense-Aspect-Mood Inflection | 25B Zero Marking of A and P Arguments | 29A Syncretism in Verbal Person/Number Marking |
|---|---|---|---|---|---|---|
| ita | 2 Strongly suffixing | 3 No productive reduplication | 3 Core and non-core | 2 TAM+agreement | 2 Non-zero marking | 2 Syncretic |
| eng | 2 Strongly suffixing | 3 No productive reduplication | 2 Core cases only | 1 monoexponential TAM | 2 Non-zero marking | 2 Syncretic |
| tur | 2 Strongly suffixing | 1 Productive full and partial reduplication | 4 No syncretism | 1 monoexponential TAM | 2 Non-zero marking | 3 Not syncretic |
| fin | 2 Strongly suffixing | 3 No productive reduplication | 3 Core and non-core | 1 monoexponential TAM | 2 Non-zero marking | 3 Not syncretic |
| swe | 2 Strongly suffixing | 3 No productive reduplication | 2 Core case only | 1 monoexponential TAM | 2 Non-zero marking | 2 Syncretic |
| prs | 3 Weakly suffixing | 1 Productive full and partial reduplication | 1 No case marking | 1 monoexponential TAM | 2 Non-zero marking | 3 Not syncretic |
| ger | 2 Strongly suffixing | 3 No productive reduplication | 3 Core and non-core | 1 monoexponential TAM | 2 Non-zero marking | 2 Syncretic |
| fre | 2 Strongly suffixing | 3 No productive reduplication | 3 Core and non-core | 2 TAM+agreement | 2 Non-zero marking | 2 Syncretic |
| rus | 2 Strongly suffixing | 3 No productive reduplication | 3 Core and non-core | 1 monoexponential TAM | 2 Non-zero marking | 3 Not syncretic |
| spa | 2 Strongly suffixing | 3 No productive reduplication | 3 Core and non-core | 2 TAM+agreement | 2 Non-zero marking | 2 Syncretic |
| dut | 2 Strongly suffixing | 3 No productive reduplication | 2 Core cases only | 1 monoexponential TAM | 2 Non-zero marking | 2 Syncretic |
| rom | 2 Strongly suffixing | 3 No productive reduplication | 3 Core and non-core | 2 TAM+agreement | 2 Non-zero marking | 2 Syncretic |
| grk | 2 Strongly suffixing | 3 No productive reduplication | 3 Core and non-core | 3 TAM+agreement+diathesis | 2 Non-zero marking | 3 Not syncretic |

## A.2 Correlation Results on Clustered languages

**Clustering of languages** As described in 5.2, a K-means clustering of languages is performed for both syntactic and morphological typological spaces. Four clusters emerged for the syntactic typological space of languages, and three clusters emerged on the morphological one.

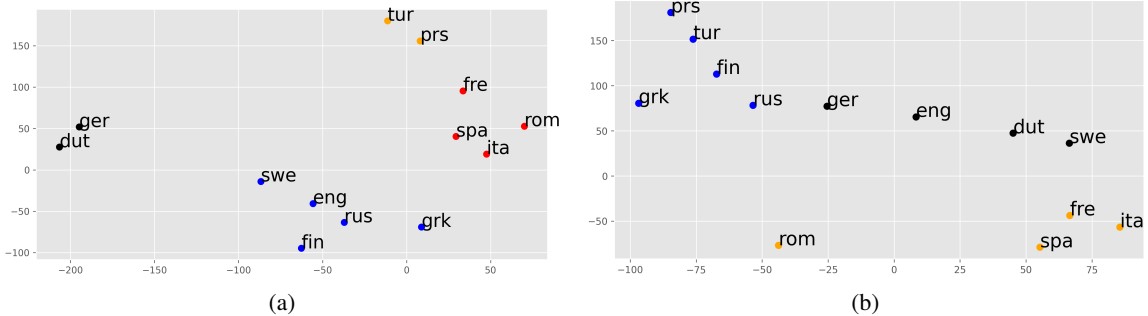

Figure 4: t-SNE plot of clustering based on syntactic (4a) and morphological (4b) features extracted from WALS.

**Results Analysis** Here we report results between all syntactic and morpholocival clusters. In Figure 6 and Figure 5, one for each matrix, two different clusters are considered.

From the syntactic point of view, as described in Section 5.2, middle layers similarities are positively correlated with typological similarities and this positive correlation can be observed also after clustering, in Figure 6a, when a sufficient number of languages is considered. Results are very unstable when one of the considered cluster is smaller (see Figure 6b - 6f). In fact, Spearman's values that exceed the threshold of $0.5$ fluctuate too much across different layers and matrices and no pattern can be hence clearly observed. Results are rarely statistically significant, with the exception of $Q_3$ in Figure 6e. Hence, it is hard to assert that the polarizing features of these clusters are encoded at some layers using Spearman's correlation coefficient.

From the morphological point of view (see Fig. 5), language pairs from $M1$-$M3$ and, marginally, from $M2$-$M3$ lead to interesting results. Indeed, related ranked lists for pairs for these clusters show extra-clustering Spearman's coefficients that are above the threshold across multiple layers and that are statistically significant. The rankings of pairs generated from $M1$-$M3$ and $M2$-$M3$ have peaks at layer 0 ($Q_0$ matrix) and layer 3 ($V_3$ matrix), respectively. However, a high correlation can also be observed in other layers, with no clear descending trend.

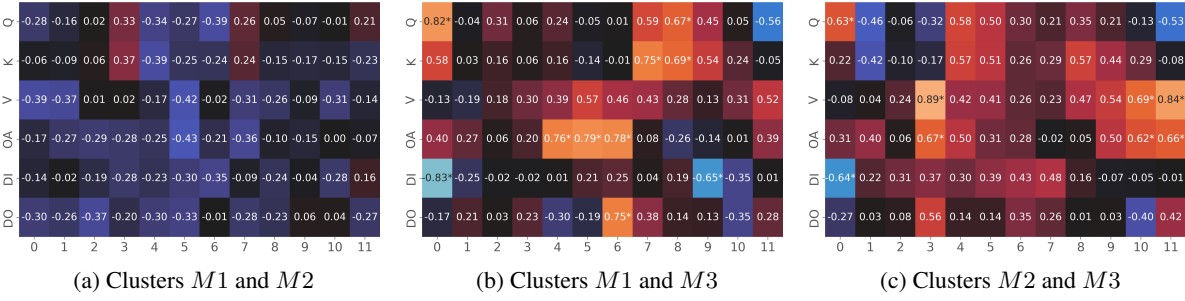

Figure 5: Each matrix shows the Spearman's correlation coefficients for extra-cluster morphological analysis, one matrix for each pair of clusters, $M_i$ and $M_j$. Values closer to $+1$ are in red, values closer to -1 in blue. Statistically significant results with a p-value lower than $0.01$ are labelled with $^*$.

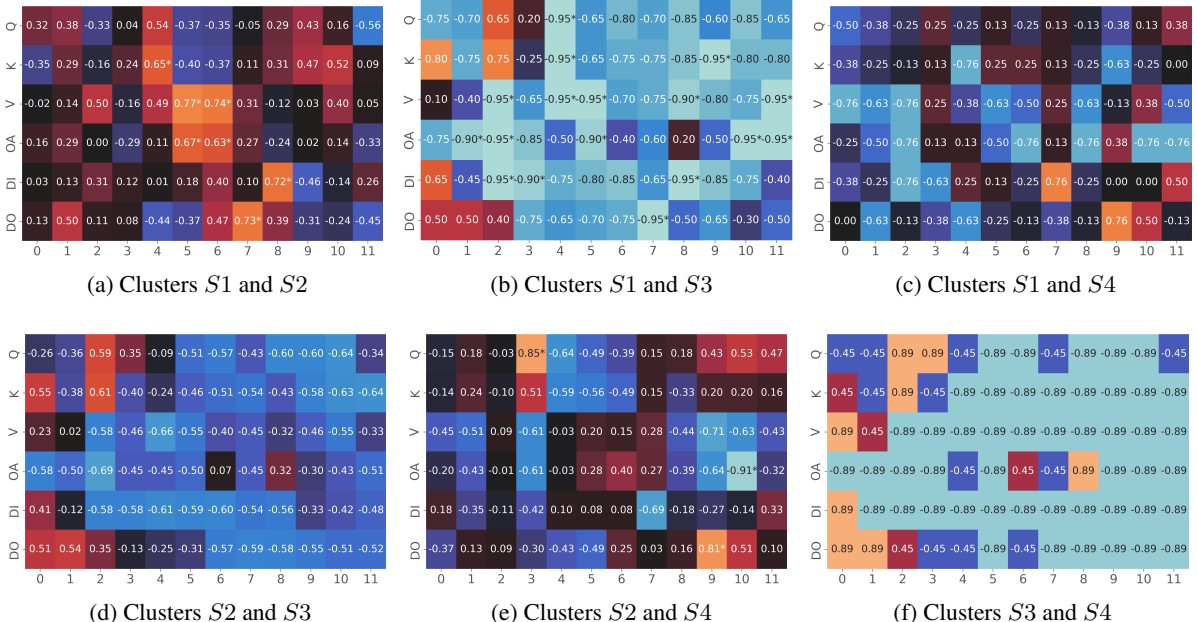

(a) Clusters $S1$ and $S2$  (b) Clusters $S1$ and $S3$  (c) Clusters $S1$ and $S4$

(d) Clusters $S2$ and $S3$  (e) Clusters $S2$ and $S4$  (f) Clusters $S3$ and $S4$

Figure 6: Each matrix shows the Spearman's correlation coefficients for extra-cluster syntactic analysis, one matrix for each pair of clusters, $S_i$ and $S_j$. Values closer to $+1$ are in red, and values closer to -1 in blue. Statistically significant results with a p-value lower than $0.01$ are labeled with $^*$.

## A.3 Correlation Results after Finetuning

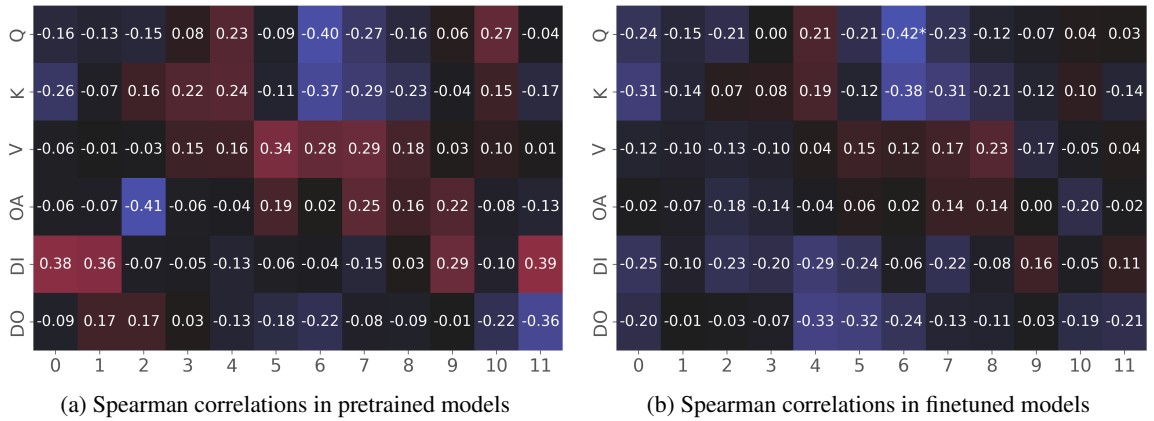

(a) Spearman correlations in pretrained models  (b) Spearman correlations in finetuned models

Figure 7

As described in Section 5.3, the typological similarities among languages has a positive correlation with the similarity observed across different models, and this correlations are more clearly observed after finetuning. With morphological analysis, conversely, we cannot draw further insights into the similarities between the different models. On this smaller set of languages, no positive statistically significant correlations can be observed either in pretrained models 7a or after finetuning 7b.