# OpenReview forum: "Exploring Linguistic Properties of Monolingual BERTs with Typological Classification among Languages"
_EMNLP/2023/Conference — EMNLP 2023 Findings_

### Official Review · Reviewer_DrBj · 2023-08-04

**Soundness:** 4

**Excitement:**

4: Strong: This paper deepens the understanding of some phenomenon or lowers the barriers to an existing research direction.

**Paper Topic And Main Contributions:**

The paper aims to validate where morphology and syntax are encoded in pre-trained monolingual BERT models. The authors leverage Centered Kernel Alignment to assess the similarity between weight matrices in BERT and conclude that indeed syntax is encoded in the middle layers of BERT and morphology is more akin to be encoded by the attention layer, though less strongly. The authors also convincingly demonstrate that syntax remains encoded in the middle layers across typologically different languages.

The main contributions of the paper are a methodology to assess where specific linguistic phenomena are encoded in pre-trained models and an experimental validation on where morphology and syntax are encoded in multiple (including on low-resource languages) pre-trained monolingual BERT models. As a consequence of their experiments, the authors arrive at a conclusion analogous to Tenney et al. (2019) on where BERT appears to encode syntactical information and thereby independently validate the earlier results with a slightly different methodological appraoch. A (side) contribution of the work is the addition of missing features in WALS which the authors leverage for their experiments.



**Questions For The Authors:**

* Line 077: Can you give an example or explain more concisely what you mean by "syntactic typological similarity"? At this point in the paper its not at all clear what exactly you are measuring.
* Line 144: You are saying "[...] mBERT is not exploiting typological similarities among languages [...]". Why would you assume that it would or should? I.e. is this enforced in the training process or the objective function?
* Line 200: Not everybody is familiar how lang2vec works, if your work is a variation of lang2vec, it would be good to briefly explain how lang2vec works, if your approach is just similar, then it makes more sense to focus on how your approach works (my reading is that this is what is happening, however the sentence "[...] we use a typological classification scheme that, in line with lang2vec [...]" got me slightly confused since I am not familiar enough with lang2vec to immediately pick up what you mean).
* Since you've completed some missing features in WALS - have you contributed this to the WALS project? If so, then this would be a very welcome (side-)contribution of the project worth mentioning.
* Line 379: You mention that you use pre-trained models in the various languages from hugginface. These models will have been trained on a variety of different corpora. Do you expect any variation due the different training corpora used?
* Line 604-608: Can you give more reasoning and insight why these findings are important (I don't mean to be cynical, I do find the results interesting _per se_, however I would also be interested in what way you would leverage these findings)?

**Reasons To Accept:**

* This is a very well-written paper, posing a clear research question and addresses this question empirically.
* Its great to see findings of earlier work by Tenney et al. (2019) confirmed by a different methodological approach.
* The experimental setup is really cool - I especially liked the use of WALS to cluster languages in combination with the FOCUSED experiments.
* A key contribution of the work is not just the findings on BERT, but the methodology as such which would be applicable to models beyond BERT (I believe this contribution could be claimed more prominently).

**Reasons To Reject:**

* I don't see any reasons for rejecting this work. It poses a clear research hypothesis and contains a sound methodological approach to validate the hypothesis.

**Reproducibility:**

4: Could mostly reproduce the results, but there may be some variation because of sample variance or minor variations in their interpretation of the protocol or method.

**Reviewer Confidence:**

4: Quite sure. I tried to check the important points carefully. It's unlikely, though conceivable, that I missed something that should affect my ratings.

**Typos Grammar Style And Presentation Improvements:**

* The first sentence in the abstract - "[...] of transformers force NLP researchers [...]" - is quite awkward, especially that we're _forced_ to delve into these models.
* Line 030: Peters et al. (2018) did not use a pre-trained transformer (ELMo was a deep biLSTM).
* Line 186: Missing "representations" in "[...] their BERT _[representations]_ should have similar weight matrices [...]"

---

> ### Author Rebuttal · Authors · 2023-08-25
>
> __Answering to Questions to the Authors:__
>
> * Line 077:
> Typological classification of languages deals with similar structural features -- syntactic and morphological -- across different languages. For example, to construct a syntactic typological classification, one feature that is taken into consideration is the basic order of constituents -- subject (S), direct object (O) and verb (V) -- in a clause. In WALS, this feature is called "81A - Order of Subject, Object and Verb".
> For example Japanese and Turkish generally share the same word order.
> A sentence like _Mary eats an apple_,
> in Japanese is translated as _Mary_ (S – ‘Mary’) _wa_ (particle) _ringo_ (O – ‘apple’) _wo_ (particle) _tabemasu_ (V – ‘eats’)
> and in Turkish is _Mary_ (S – 'Mary’) _elmayi_ (O – ‘the apple’) _yiyor_ (V – ‘eats’).
> This is not the case in Italian (and in English), since the order is different _Mary_ (S – ‘Mary’) _mangia_ (V – ‘eats’) _la mela_ (O – ‘the apple’).
>
>
> * Line 144:
> We stem from Singh et al.(2019) findings. In fact, since the representations produced by mBERT seem to be separated between different languages, this model does not exploit similarity among languages to reuse part of the representation it builds across different languages.
>
> * Line 200:
> Our approach is definitely similar to lang2vec, since it defines a list of features for each language and also contains some WALS features.
> It also defines language vector as one-hot encoding of those features and defines their similarities via cosine similarity.
> However, lang2vec is not complete (morphological features are not defined). Hence, we started from WALS and we add some feature values that were missing.
>
> * We could have emphasized  more that our side contribution is an extension of WALS! We plan to contribute to the WALS project with the actual missing values that we provided.
>
> * Line 379:
> Selecting models from huggingface we observed that they were all pretrained on a large variety of sources and domains.
> However, dissimilar pretraining data could negatively affect the analyzed correlations.
> We tried to understand what could happen when more similar sources are employed in Section 5.3. As future work, we would like to analyze on more similar models and expect to get higher similarities for more similar languages. In this phase, we chose to include as large number of languages as possible to make the findings more generalizable.
>
> * Line 604-608:
> These findings can be important because they can help interpret the inner workings of these models to go toward the so-called actionable explainability - the idea of giving the possibility to act within the matrices changing some weights to obtain a desired behavior. Moreover, this methodology could help to build a more informed architecture that takes advantage of the most promising layers in multilingual models. We will add these considerations in the conclusions.

---

### Official Review · Reviewer_WpkQ · 2023-08-04

**Typos Grammar Style And Presentation Improvements:** N/A
**Soundness:** 3

**Excitement:**

3: Ambivalent: It has merits (e.g., it reports state-of-the-art results, the idea is nice), but there are key weaknesses (e.g., it describes incremental work), and it can significantly benefit from another round of revision. However, I won't object to accepting it if my co-reviewers champion it.

**Missing References:**

N/A

**Paper Topic And Main Contributions:**

This paper tried to investigate how the transformers based models represent the underlying structure of languages. The main intuition of this paper is: if different natural languages are syntactically or morphologically similar then their BERT should have similar weight matrices in some layers. According the experimental results, layers 4,5 and 6 encode syntactic information of languages.

**Questions For The Authors:**

What are the significances of your research (opening a new way to investigate specific linguistic features in layers.)?
How did you extract or feed the typological features into the BERT model?

**Reasons To Accept:**

The authors tried to investigate a new property of transformer-based models and the results confirm their claims.

**Reasons To Reject:**

Though the authors discovered a novel property of transformer models, I think the main limitation of this paper is they didn't discuss about how they represent the syntactic and morphological features of any languages as vectors (there is no explanation of lang2vec) or how a pretrained BERT for a language L, weight matrices Wi = {Qi,Ki, Vi,OAi,DIi,DOi} represent the typological features of a language or how they extract the syntactic and morphological features from a BERT model. According to my understanding, they just measured the similarities among weight matrices.

**Reproducibility:**

3: Could reproduce the results with some difficulty. The settings of parameters are underspecified or subjectively determined; the training/evaluation data are not widely available.

**Reviewer Confidence:**

3: Pretty sure, but there's a chance I missed something. Although I have a good feel for this area in general, I did not carefully check the paper's details, e.g., the math, experimental design, or novelty.

---

> ### Author Rebuttal · Authors · 2023-08-25
>
> __Commenting on Reasons to Reject and Answering to Questions to the Authors:__
>
> _Raised point: Significance of research and methodology_
>
> Opening a new way to investigate specific linguistic features in Transformers layers is exactly our purpose: we propose a new methodology to investigate specific linguistic features in layers based on comparing models of different languages to study whether linguistic similarities between languages are reflected in similarities among matrices in specific layers! Hence, we are not encoding syntax and morphology. We are detecting where it is encoded in a new way.
>
> As required, we will better clarify at the end of Section 3.1 how we define language feature vectors and the similarity among them. Just a few missing lines: We produce the one-hot encoding of features described by linguists in WALS for each language and then compute the cosine similarity between them to get assessments of similarities.
>
> With our new methodology, we discover that models' weight matrices carry important information about the languages they were trained on, as they encode syntactic and morphological features of languages in specific layers.
>
> The core idea is that, if the same similarity that holds between languages (comparing typological features) also holds between models (comparing weight matrices), it means that the investigated BERT models encode typological features of languages.
>
> For this purpose, we use biCKA to measure the similarity between weight matrices, and we use the cosine similarity between language feature vectors to measure typological similarity between languages. Then we correlate these two assessments of similarities with Spearman Correlation Coefficient. Hence, to define the ability of monolingual models to retain typological information, we do not need to extract typological features from weight matrices.

---

### Official Review · Reviewer_j8oq · 2023-08-05

**Typos Grammar Style And Presentation Improvements:** L234
**Soundness:** 3

**Excitement:**

3: Ambivalent: It has merits (e.g., it reports state-of-the-art results, the idea is nice), but there are key weaknesses (e.g., it describes incremental work), and it can significantly benefit from another round of revision. However, I won't object to accepting it if my co-reviewers champion it.

**Paper Topic And Main Contributions:**

This paper investigates how typological linguistic features for syntax and morphology are encoded in monolingual models, then compares different monolingual builds between typologically similar/dissimilar languages to gain insight into shared properties. The comparison is done on the weight matrices for model layers and results in showing similar activations between models with shared linguistic features. The paper contributes using bidimensional centered kernel alignment to compare activation matrices across both rows and columns. The paper overall contributes to methods of using linguistically informed decisions to make more motivated model build decisions.

**Questions For The Authors:**

QuestionA: How much of an impact do you think using the monolingual models that were trained on a variety of different data sources makes an impact of the findings here? Should the models not be controlled for in any way?

**Reasons To Accept:**

- Linguistically motivated analysis that considers similarities between languages and their monolingual model build; may help others in determining language combinations for multilingual models or for doing pretraining with multiple languages for one language
- It is important to be able to analyze what kind of linguistic information gets encoded and used in model builds so the paper is a start for this understanding


**Reasons To Reject:**

- It would have been interesting to see more detailed discussion of which linguistic features shared between the languages had more or less impact on the matrix activations as opposed to other dissimilar languages; that is, more of an ablation study on the different activations
- Additional discussion on why certain features between which languages were activated may illustrate more how to use this findings in other experimental set ups - can/how this be extended to multilingual builds for example?

**Reproducibility:**

3: Could reproduce the results with some difficulty. The settings of parameters are underspecified or subjectively determined; the training/evaluation data are not widely available.

**Reviewer Confidence:**

3: Pretty sure, but there's a chance I missed something. Although I have a good feel for this area in general, I did not carefully check the paper's details, e.g., the math, experimental design, or novelty.

---

> ### Author Rebuttal · Authors · 2023-08-25
>
> __Commenting on Reasons to Reject:__
>
> _Raised point: Discussion of linguistic features_
>
> Indeed! In Section 5.2., we partially addressed the issue, but we need to make it a little bit more explicit. Understanding which linguistic features shared between the languages have a larger impact on similarities between matrices is interesting.
>
> Just to clarify. In Section 5.2, we identified linguistic features that make languages more different, and we named them "polarizing'' (meaning that they assume the same value within one cluster and the opposite value in the other). This is also why we later refer to them as important (L234), because they make clusters distinguishable.  Definitely, we need to add some words to state this point more clearly. Yet, the idea is that those matrices could be responsible for encoding information about the relative order of words since the polarizing features we detected ("87A Order of Adjective and Noun'' and "97A Relationship between the Order of Object and Verb and the Order of Adjective and Noun") mainly refer to this kind of syntactical information.
>
> This kind of information could provide insights, for example, into more effective multilingual training in languages that share the same word order. However, this analysis is meaningful only when large enough clusters are available, and hence, we commented only on these features for the larger pair of syntactic clusters.
>
> __Answering to Questions to the Authors:__
>
> _Question A: Variety of data sources_
>
> The great variety of pretraining data can, in fact, negatively affect the analyzed correlations. We initially considered analyzing only models exposed to similar data sources. However, this would eventually lead to a smaller number of models and languages. Moreover, it could make the investigation less general.
> Hence, on the one hand, we introduced a single, looser constraint and chose to include only BERT models pre-trained on a single language and with sufficiently large and documented datasets. On the other hand, we decided to have an intuition of what could happen when models are exposed to more similar sources (in Section 5.3), even if the setting can only give an intuition of what can happen when models are trained from scratch on more similar (or parallel) data sources. We will state this point more clearly!

---

### Meta-Review · Area_Chair_GFj7 · 2023-09-13

**Recommendation:** 4

**Metareview:**

The paper aims to validate where morphology and syntax are encoded in pre-trained monolingual BERT models. The authors leverage Centered Kernel Alignment to assess the similarity between weight matrices in BERT and conclude that indeed syntax is encoded in the middle layers of BERT and morphology is more akin to be encoded by the attention layer, though less strongly. The authors also convincingly demonstrate that syntax remains encoded in the middle layers across typologically different languages.

The main contributions of the paper are a methodology to assess where specific linguistic phenomena are encoded in pre-trained models and an experimental validation on where morphology and syntax are encoded in multiple (including on low-resource languages) pre-trained monolingual BERT models. As a consequence of their experiments, the authors arrive at a conclusion analogous to Tenney et al. (2019) on where BERT appears to encode syntactical information and thereby independently validate the earlier results with a slightly different methodological appraoch. A (side) contribution of the work is the addition of missing features in WALS which the authors leverage for their experiments.

All in all the paper is in a good shape. Some additional discussion and anaylsis may be added but this is not a major issue.

---

### Decision · Program_Chairs · 2023-10-07

**Decision:**

Accept-Findings

**Comment:**

The paper aims to validate where morphology and syntax are encoded in pre-trained monolingual BERT models. The authors leverage Centered Kernel Alignment to assess the similarity between weight matrices in BERT and conclude that indeed syntax is encoded in the middle layers of BERT and morphology is more akin to be encoded by the attention layer, though less strongly. The authors also convincingly demonstrate that syntax remains encoded in the middle layers across typologically different languages.

The main contributions of the paper are a methodology to assess where specific linguistic phenomena are encoded in pre-trained models and an experimental validation on where morphology and syntax are encoded in multiple (including on low-resource languages) pre-trained monolingual BERT models. As a consequence of their experiments, the authors arrive at a conclusion analogous to Tenney et al. (2019) on where BERT appears to encode syntactical information and thereby independently validate the earlier results with a slightly different methodological appraoch. A (side) contribution of the work is the addition of missing features in WALS which the authors leverage for their experiments.

All in all the paper is in a good shape. Some additional discussion and anaylsis may be added but this is not a major issue.